# Prioritization-Driven Congestion Control in Networks for the Internet of Medical Things: A Cross-Layer Proposal

**DOI:** 10.3390/s23020923

**Published:** 2023-01-13

**Authors:** Raymundo Buenrostro-Mariscal, Pedro C. Santana-Mancilla, Osval Antonio Montesinos-López, Mabel Vazquez-Briseno, Juan Ivan Nieto-Hipolito

**Affiliations:** 1School of Telematics, Universidad de Colima, Colima 28040, Mexico; 2Facultad de Ingeniería, Arquitectura y Diseño, Universidad Autónoma de Baja California, Ensenada 22860, Mexico

**Keywords:** congestion control, packet scheduling, cross-layer, healthcare, internet of medical things

## Abstract

Real-life implementation of the Internet of Things (IoT) in healthcare requires sufficient quality of service (QoS) to transmit the collected data successfully. However, unsolved challenges in prioritization and congestion issues limit the functionality of IoT networks by increasing the likelihood of packet loss, latency, and high-power consumption in healthcare systems. This study proposes a priority-based cross-layer congestion control protocol called QCCP, which is managed by communication devices’ transport and medium access control (MAC) layers. Unlike existing methods, the novelty of QCCP is how it estimates and resolves wireless channel congestion because it does not generate control packets, operates in a distributed manner, and only has a one-bit overhead. Furthermore, at the same time, QCCP offers packet scheduling considering each packet’s network load and QoS. The results of the experiments demonstrated that with a 95% confidence level, QCCP achieves sufficient performance to support the QoS requirements for the transmission of health signals. Finally, the comparison study shows that QCCP outperforms other TCP protocols, with 64.31% higher throughput, 18.66% less packet loss, and 47.87% less latency.

## 1. Introduction

Owing to the emergence and increasing adoption of the Internet of Things (IoT), society is increasingly moving towards an always-connected model, where there are smart objects that interact with each other and with people [1,2]. IoT offers suitable solutions for numerous applications, such as smart cities, security, emergency services, logistics, commerce, industrial control, and healthcare [3,4,5]. Healthcare is considered one of the areas that will benefit the most from IoT, which has led to the coining of the term Internet of Medical Things (IoMT). IoMT [6,7] includes devices with multiple sensors placed on, near, or inside a patient to collect vital sign data and wirelessly transmit it to the sink node via hop-by-hop (see Figure 1). In turn, the sink node connects to the Internet to send the data to an application that can process the data to detect abnormalities in a patient’s health [8]. However, several technical challenges must be overcome before IoT technologies can be implemented in healthcare settings. Providing quality of service (QoS) is one of the most significant challenges [9,10,11,12,13,14]. Furthermore, considering the use of wireless networks and the presence of network congestion, this challenge becomes more difficult to solve [9,15]. For example, various healthcare applications that send and receive electrocardiogram (ECG) data can cause a significant increase in data traffic, which can lead to network congestion. Congestion plays a significant role in packet loss, forwarding delay, and node power consumption and negatively impacts QoS in an IoMT environment [16,17]. Furthermore, packet loss and latency increase significantly when the source node (node including sensors) is several hops away from the target node. Therefore, it is important to integrate intelligent protocols that handle traffic prioritization and congestion control in IoMT environments [18]. Several studies have proposed solutions for the problem of network congestion. However, these solutions are complex and involve modifying the specifications of the implicated standards, which makes them impractical, particularly for IoT devices with limited computing resources [19]. Furthermore, few studies have considered cross-layer strategies for optimizing congestion control. Cross-layer strategies improve the performance of communication protocols because they are characterized by information sharing between different layers [14,20,21,22,23]. Motivated by the limitations of the existing protocols, this study proposes a novel prioritization-based cross-layer congestion control protocol called QCCP. In our proposal, QCCP is embedded both in the source and intermediate nodes, as well as in the sink node of Figure 1. In addition, each node operates with a lightweight application layer for sensing physiological variables, a transport layer to send the collected information to the sink node, and a network layer for forwarding functions. QCCP is proposed to operate at the transport layer for congestion control and packet prioritization functions. Finally, the sink node has the necessary resources to send the information to the Internet.

QCCP processes all tasks using four significant functions: packet prioritization, congestion detection, congestion notification, and congestion resolution. The results obtained in the tests showed that QCCP is a solution that can offer traffic priority while achieving adequate performance in highly congested networks. The main advantages and contributions of our proposal are as follows:QCCP is a multi-objective protocol since congestion control and prioritization policies are adjusted based on various performance objectives, such as latency, packet loss, and node balance.QCCP supports multiple applications with different performance demands simultaneously. This is possible because QCCP abstracts from particular applications, and instead categorizes them into three classes of services: urgent (P1), important (P2), and best effort (P3).QCCP does not require complementary software or agents for its operation, unlike other proposals that will be detailed in Section 2. Furthermore, it does not need to modify the standard protocols of the lower layers of the node.QCCP proposes a packet scheduler that interacts with the node’s medium access control (MAC) sublayer to work synchronously on packet prioritization and congestion control.QCCP followed design principles to produce an efficient and lightweight protocol, such as: not generating control packets, minimum overhead (one bit), TCP/IP compatibility, decentralized operation, and minimum requirements of computing and power consumption.

To better understand this problem, the entire document is organized as follows. In Section 2, we present related works. In Section 3, we explain the network and node models. Section 4 describes the modules and functionalities of the QCCP. The experiments are presented in Section 5. In Section 6, we explain the experimental results of the proposed scheme. Section 7 presents a statistical analysis to validate our results. Finally, the paper is concluded, and future research directions are provided in Section 8.

## 2. Related Works

Most existing strategies for solving congestion are based on the transmission regulation rate of nodes or multiple routes that can be developed hop-by-hop (open loop) or end-to-end (closed loop). One proposal is HOCA [17], which uses a buffer-usage threshold. HOCA employs explicit notifications to broadcast congestion alerts across the network and defines new routes to forward data if the node has a routing algorithm. In addition, it executes rate adjustment in the source code. Its limitations include the need for a routing algorithm, centralized control, and increased network overhead owing to explicit messages. The authors of [24] proposed an in-node data aggregation technique that eliminates redundancy in sensed data to prevent network congestion (called EECCDD). This technique consists of partially processing the data at each member node and forwarding a fraction of the actual data, that is, the fused data, towards the cluster head. Consequently, communication costs, packet collisions, and network congestion are reduced, and the network lifetime is enhanced. However, despite this exciting idea, its applicability is reduced to scenarios with nodes that produce data with high correlation. A socially aware congestion control algorithm (SACC) addresses the congestion problem using the perspective of socially aware computing for delay-tolerant networks (DTN) [25]. SACC uses the social features of a network and the congestion level of a node to build a social congestion metric (SCM) using a novel message-dropping mechanism. When congestion occurs, the node calculates its social links (SL) with the destination node of each message and then drops the message with the minimum SL to eliminate congestion. However, it is likely that the historical value of frequent contact between nodes, or SL, defines a reliable route at a specific time to transmit a message. Additionally, for the selective dropping of messages, the SACC considers the social relationship between nodes instead of the importance of a message and is not QoS-aware. The authors of [26] proposed a rate-control protocol called PCC-Vivace based on machine learning, which operates up the transport layer. Their simulation results showed that their protocol outperforms better than traditional TCP variants. However, PCC-Vivace requires a middleware located at the side of the sender, which implies having a high compute resource source node. This is the main difference with our approach, which is based on a wireless sensor network, characterized by its scarcity of computing resources such as memory and computing capacity. The authors of [27] propose a multi-objective congestion control (MOCC) algorithm that is based on a multi-objective reinforcement learning (RL) framework, which automatically learns the correlations between the different application requirements and the corresponding optimal control policies. The design of this framework uses a policy neural network and a preference neural sub-network, with three key functions, such as the registration of application requirements; obtaining the latest network conditions; and defining the sending rate for packet transmission accordingly. For better portability, MOCC functions have been encapsulated in a library that can be used by other algorithms to control network congestion. In their reported implementation, the authors used the UDT protocol [28], similar to UDP, and the CCP protocol [29] for congestion control and to feed the MOCC kernel with network status. In the results of its simulation tests, the solution offers better performance than other protocols, in terms of throughput and latency, adaptation to new applications and high bitrate to handle video streaming over the Internet. This proposal, however, has some drawbacks, such as the need to use other protocols, such as UDT and CCP, to complete the three functions of MOCC. In addition, MOCC uses neural networks and reinforcement learning, which can lead to a complex solution that demands more computing and storage capabilities. Regarding congestion control, MOCC is based on the adjustment of the packet transmission rate and does not incorporate other control actions, such as packet prioritization. Therefore, these points must be analyzed when selecting a solution for IoMT networks. The priority-based congestion control protocol (PCCP) is proposed by [30] to manage congestion in a sensor network. PCCP determines the degree of congestion using the packet interarrival time and the packet service time in the MAC sublayer. To resolve the congestion, PCCP implements a rate adjustment function in the node, according to the degree of congestion and the priority indices of the nodes. To compute this adjustment, PCCP needs to notify the congestion degree to the nodes in the neighborhood, so that all nodes can calculate the global and local priority of the node and act accordingly. This procedure helps to improve the estimate of global network congestion; however, at the same time, it becomes a weakness since the information from other nodes is needed to complete the congestion control operation. The work reports good results in the performance of the PCCP protocol against different degrees of network congestion. However, there are some PCCP design weaknesses that must be considered if it is to be used in practice. PCCP performs the priority only for local and external traffic of the node, but it does not distinguish the traffic by type of application of higher layers, leaving the prioritization function very limited. In addition, the authors made a biased evaluation since they assume impractical aspects, for example, a MAC protocol that provides uniform access opportunities for each node assumes that each sensor node has the same source traffic priority index, and that the node sink could obtain the number of packets from each sensor node. With the intention of ensuring high throughput and low packet losses, the authors propose the Dynamic-LIA (D-LIA) congestion control protocol [31], which is an enhancement of the linked-increases algorithm (LIA) [32]. The improvement lies in executing the congestion window (CWND) reduction in a better way each time there is a packet loss. LIA executes this action aggressively, halving the CWND like traditional TCP. For its part, D-LIA decreases the CWND by a factor, which determines it dynamically based on the value of the time interval between packet loss events, reaching an optimal CWND value quickly and using the available bandwidth appropriately. This change allows D-LIA to detect a sudden change in the network, thus avoiding aggressive CWND reduction. In their results, they show that the performance of D-LIA is superior to other congestion control protocols, mainly in terms of throughput and fairness. Among its areas of opportunity are the following: D-LIA considers packet loss as the only cause of network congestion; however, this form is very limited, especially in wireless networks where packet loss can be caused by channel noise. Another disadvantage of D-LIA, which the authors acknowledge, is the increased packet retransmission compared to the LIA operation. In addition, the results obtained in the tests are limited, since their test scenarios have only two pieces of terminal equipment and only two paths between them, which is not enough for a real and highly congested scenario. The authors propose a modified Fast-Vegas-LIA hybrid congestion control algorithm (MFVLHCCA) [33], which considers some attributes of the Fast-TCP [34], modified TCP Vegas [35], and LIA [32] algorithms. MFVLHCCA was designed to improve congestion control of the Multipath-TCP protocol (MPTCP, designed by the IETF to support multiple paths for a TCP session in multihomed networks). MFVLHCCA can operate in either uncoupled (where adjusts the CWND of a flow without considering other flows on the same path) or coupled congestion control mode. It uses the decoupled congestion control mode if there is no shared bottleneck, otherwise, it switches to the coupled congestion control mode just like the LIA algorithm. Working both modes achieves good throughput and fairness for other flows within the route, especially when there is a non-shared bottleneck. MFVLHCCA was compared to other congestion control algorithms, and test results in the decoupled mode show a 50% reduction in packet loss and a 30% increase in average goodput. One of the weaknesses of MFVLHCCA is that it assumes that all flows in the node experience the same network latency, otherwise the timing of the algorithm is a problem; however, in fact, the flows in the network generally have different latencies. Another assumption of MFVLHCCA is that the IoT network has devices equipped with high-speed processors, large on-chip memories, and multiple network interfaces, which is not common nowadays and less in IoMT networks.

Despite various studies on congested network problems, challenges still need to be addressed in order to provide efficient, lightweight, and practical solutions.

### Motivation for Congestion Control in IoMT Networks

The main differences between QCCP and the reviewed works are given below, with the intention of showing the opportunity areas addressed by our proposal. One of these differences is the number of metrics used to estimate node congestion. For example, QCCP uses the metrics of node congestion degree (C), the number of packets dropped due to channel access failures (CAF), and packet processing delay (D_P_). Unlike HOCA and DALIA, which use only one metric, either buffer usage or packet loss, respectively. QCCP uses two mechanisms to resolve congestion (packet scheduling and medium access management), unlike SACC which uses only packet dropping, and MOCC and PCCP which use the node’s transmission rate adjustment. Another important aspect is that QCCP does not need other algorithms, agents, or information from other nodes to execute its substantive functions; unlike HOCA, MOCC, PCCP, EECCDD, and PCC-Vivace, which require a routing algorithm, a transport agent, information from other nodes, or un middleware in sensor nodes, respectively. The SACC, PCCP, DLIA, and MFVLHCCA proposals do not consider the priority of the packets (according to the type of application) to perform congestion control, which is a major weakness. For this reason, QCCP considers the priority of each packet, defined by the application layer, as the primary axis in decision-making for all its key processes. QCCP is very clear that the solution should not have a high overhead, so as not to contribute to congestion and detract from the network usefulness. In this regard, HOCA, MFVLHCCA, and SACC present solutions with a large overhead; unlike QCCP which uses a one-bit overhead. In the works reviewed, it was observed that some proposals require network devices with complex hardware and extensive computing and storage capacities. For example, MOCC uses a neural network algorithm and reinforcement learning, PCC-Vivace uses a machine learning algorithm, SACC requires a large space to store a series of state tables, and MFVLHCCA requires devices with multiple network interfaces. QCCP, on the other hand, is designed based on limited resource devices, as defined by the IEEE 802.15.4 standard [36], which are intended for networks with simple processes, and minimal hardware and software capabilities. In conclusion, QCCP has several significant differences to the cited works, which position QCCP as a traffic prioritization and congestion control solution that can be considered for IoMT networks.

## 3. System Model

### 3.1. Network Model

Data acquisition from medical sensors is the first task realized by an IoMT, so the network model considered in this study was formed by a set of IoMT nodes grouped into clusters based on link quality proximity (Figure 2). Our IoMT network is formed by three kinds of nodes: (i) Source node, responsible for generating data and depending on its sensors providing the correspondence type of traffic at r_loc_ rate. In Figure 2, they are labeled as 4, 5, and 6 and they forward their data to an intermediate node located in their range of operation. (ii) The intermediate node is responsible for generating its own traffic (if necessary) and to relay traffic to the next node up to the sink node. Intermediate nodes, which are in the path between the source nodes and sink node (for example, node 1), generate traffic at $\sum^{\;}r_{\mathrm{tr}}+r_{\mathrm{loc}}$ rate; where r_loc_ is its local traffic (if there were), and r_tr_ is the traffic it receives from other nodes behind it, either source nodes or other intermediate nodes. Consequently, intermediate nodes have a higher traffic relaying burden than source nodes. In Figure 2, they are labeled as 1, 2, and 3. (iii) The sink node is at the top level of the structure and is the node that receives all the traffic generated by the IoMT network, which at this point is $n(\sum r_{\mathrm{tr}}\left(i\right)+r_{\mathrm{loc}}\left(i\right))$, where n is the number of intermediate nodes and (i) is the intermediate node. Moreover, the sink node is responsible for connecting the IoMT with the Internet. The devices beyond the sink node are not part of the proposal of this work; however, an Internet network that uses a transport protocol such as TCP is assumed. In our network model, every source node can collect and transfer data to the next hop until it reaches the sink node. Therefore, the data flow of our network model is considered many-to-one convergent traffic in the upstream direction. Moreover, the application traffic is grouped into three priority levels: urgent traffic (P1); important (P2); and best effort (P3). This classification can cover the transmission requirements of most of the typical physiological signals in an IoMT network [37]. For example, the ECG signal is traffic P1 (with a data rate transmission of 15,000 bps), the heart rate signal is P2 (with a data rate transmission of 600 bps), and body temperature is P3, which is the one with the lowest transmission requirements (with a data transmission rate of 80 bps) [34].

### 3.2. Node Model

QCCP proposes a cross-layer interaction strategy (represented by dotted lines in (see Figure 3), between the transport layer and the medium access sublayer to implement its congestion control and traffic prioritization functions. QCCP is embedded in the transport layer of all nodes in the IoMT network (source, intermediate, and sink nodes). The above is possible, since all IoMT nodes have the application, transport, network, medium access, and physical layers, as explained in Section 1. QCCP interacts with the MAC sublayer to obtain information about the congestion status of the wireless channel and to set the most appropriate medium access algorithm configuration. In addition, the QCCP can connect to an application-layer protocol to inform the channel of its status (work out of scope). In the model, r_in_ represents the total traffic rate in the node created by r_loc_ (local traffic) and r_tr_ (transit traffic), which are sent to the prioritization module for processing. Subsequently, the traffic is sent to the MAC sublayer at r_prog_ rate according to the priority level of each packet. The application layer assigns a priority tag to each packet based on its QoS requirements. The MAC sublayer transmission rate (r_MAC_) is determined by the effectiveness of the medium access algorithm in placing packets in a wireless channel. 

Considering all conditions in Figure 3, the node has a service rate (r_out_) linked to r_in_ and r_MAC_, as shown in (1).
(1)$$r_{\mathrm{out}}=\{\begin{array}{cc} r_{\mathrm{MAC}} & {\mathrm{if}\;r}_{\mathrm{in}}\geq r_{\mathrm{MAC}}\\ r_{\mathrm{in}} & {\mathrm{if}\;r}_{\mathrm{in}}<r_{\mathrm{MAC}} \end{array}$$

From (1), it follows that when r_in_ < r_MAC_, the packets are served without queue delay. However, if r_in_ > r_MAC_, the node experiences congestion (or service imbalance), and the packets of the prioritization module are queued, considering queue size restrictions and queuing rules. This condition could be due to a slow packet transmission mechanism and/or congested channel. In both cases, packet loss may occur either in the prioritization module because of buffer overflow or at the MAC level because of channel access failure (CAF). Subsequently, according to the congestion levels, the QCCP protocol can reduce or increase the node output rate (r_out_). This action is carried out on the fly through the configuration of the prioritization module (r_prog_) and the medium access algorithm (r_MAC_).

The following section illustrates the rationale for the QCCP protocol and explains the modules responsible for its operations and interactions.

## 4. The Proposed Cross-Layer Scheme: QCCP Protocol

To manage heterogeneous traffic with different prioritization requirements and offer congestion control in an IoMT network, we proposed a novel priority-based cross-layer congestion control protocol (QCCP). QCCP resides in the transport layer (see Figure 3) and connects with the MAC sublayer through the service access points (SAP) defined by the IEEE 802.15.4 2006a standard [36]. Through this cross-layer connection, QCCP obtains channel traffic conditions around an IoMT device to estimate the congestion level in its neighborhood and sets up its operation accordingly. QCCP works alongside IoT application protocols such as message queuing telemetry transport (MQTT) or the constrained application protocol (CoAP) to complete the communication stack. As shown in Figure 3, QCCP has two modules: a prioritization module and a congestion control module.

### 4.1. Prioritization Module

Three components were proposed for the prioritization module: packet classifier, storage block, and packet scheduler (Figure 3). This module was designed with the primary objective of offering priority-based packet transmission services considering the current congestion level inside and around the node. In each round, this module selects which packets should be served immediately and which should be queued. Therefore, QCCP always guarantees the lowest latency (best QoS) for the most essential packages.

The process of the prioritization module starts with the classification of the packets in their corresponding buffers (packet classifier) based on packet priority tags. The storage block is then divided into buffers of different priorities (Q_1_, Q_2_, …, Q_n_), according to their QoS requirements, where buffer Q_1_ receives the highest priority traffic and Q_n_ obtains the lowest priority traffic. Finally, the prioritization module contains a packet scheduler, which is a key component of the QCCP prioritization process.

For the packet scheduler, we developed a novel congestion-aware packet service algorithm (CASPA) based on the weighted round-robin algorithm, which simply handles several priority levels and achieves a high response speed [38]. CASPA takes n packets from each buffer in rotating order according to the set weight and drives them to the MAC sublayer with an output rate of r_prog_. Furthermore, through the connection between the prioritization and congestion control modules (Figure 3), CASPA knows the current state of network saturation, allowing it to configure the weights of each buffer. With this versatility, the QCCP ensures that more packets of the most critical traffic can take the communication channel in each round. The fundamental operations followed by the CASPA algorithm are:Step 1CASPA obtains the weight values (W1, W2, and W3) for each buffer (Q1, Q2, y Q3), which are set by the congestion control module of the node.Step 2CASPA checks if there are packets in buffer Q1. If there is, dequeue a packet and decrement W1 by 1. Otherwise, it turns to step 4.Step 3CASPA checks if W1 ≠ 0. If so, it returns to step 2. Otherwise, it goes to the next buffer.Step 4CASPA checks if there are packets in buffer Q2. If there is, dequeue a packet and decrement W2 by 1. Otherwise, it turns to step 6.Step 5CASPA checks if W2 ≠ 0. If so, it returns to step 4. Otherwise, it goes to the next buffer.Step 6CASPA checks if there are packets in buffer Q3. If there is, dequeue a packet and decrement W3 by 1. Otherwise, it turns to step 1.Step 7CASPA checks if W3 ≠ 0. If so, it returns to step 6. Otherwise, it turns to step 1.

### 4.2. Congestion Control Module

This module includes three mechanisms (Figure 3): congestion detection, congestion notification, and congestion resolution. The first mechanism calculates the congestion level experienced by the node both internally (buffer overflow) and from the surrounding wireless channels (link congestion). Consequently, through a cross-layer connection, the congestion resolution mechanism sets appropriate values for the prioritization module and MAC sublayer access algorithm to prioritize the transmission of the most critical packets and suppress congestion. Finally, the congestion notification mechanism informs nodes in the neighborhood of critical network congestion (State III) by embedding a warning label within the data packet.

#### 4.2.1. Congestion Detection Mechanism

Another crucial point is that the QCCP protocol incorporates QoS parameters (delay and lost packets) to compute the degree of congestion to satisfy design objectives. To accomplish this, the congestion-detection mechanism frequently measures three parameters within a node:

##### Congestion Degree (c)

The central concept for measuring the degree of node congestion is based on (2), which compares the number of packets received by the node (r_in_) and the number of packets that it can transmit to the wireless channel (r_MAC_), as shown in Figure 3.
(2)$$C=\frac{r_{\mathrm{in}}}{r_{\mathrm{MAC}}}$$

When r_in_ is greater than r_MAC_ than C > 1, the node loses its balance and becomes congested. Depending on the calculated value of C, the mechanism identifies the node congestion state (low, medium, or high) and determines what to do next. However, (2) does not establish a direct relationship with traffic QoS requirements. Therefore, we incorporate packet processing delay (D_P_) and packet loss owing to channel access failure (CAF) to jointly establish the three congestion states.

##### Packet Processing Delay (D_p_)

The average delay time from the moment the packet arrives at the transport layer (t_in_) until the MAC sublayer transmits the same packet to the wireless channel (t_out_).
(3)$$D_{P}=t_{\mathrm{out}}-t_{\mathrm{in}}$$

##### Packet Loss by Channel Access Failure (CAF)

The average portion of packets is dropped by the MAC sublayer because of the busy channel.
(4)$$\mathrm{CAF}=\frac{\mathrm{Packet}\;\mathrm{lost}\;\mathrm{by}\;\mathrm{CAF}}{\mathrm{Packets}\;\mathrm{received}\;\mathrm{in}\;\mathrm{MAC}}*100$$

All values measured in the MAC sublayer are passed to the QCCP protocol via the cross-layer connection to execute (2)–(4). Using these three equations, we can establish the three congestion states of the node and measure their effects on QoS parameters at each moment.

#### 4.2.2. Parameter Tuning

The objective of the previous section was to define three node congestion states: low (State I), medium (State II), and high (State III). For this purpose, the average values of C, CAF, and D_P_ were measured for different traffic loads in a wireless channel. Considering the importance of these values, we propose using each metric’s exponential weighted moving average (EWMA) to avoid a poor estimate, because EWMA avoids temporary fluctuations in measurements [39].
(5)$$z_{i}={\mathrm{ax}}_{i}+\left(1-\alpha \right)z_{i-1}$$
where α is a weighting constant between 0 and 1 and is used to smooth the measured value (α = 0.1) [30]. $z_{i}$ denotes the weighted average of the ith observation. $z_{i-1}$ represents the weighted average of the last evaluation period and $x_{i}$ is the current value measured in each time interval [30]. With (5) the weighted averages of C (2), D_P_ (3), and CAF (4) were obtained.

For this tuning, a scenario comprising an IoMT with 20 sensor nodes and a sink node was used. The nodes operate on the same wireless channel and are deployed in such a way that there is full connectivity between them (the overhearing effect). First, an IEEE 802.15.4 wireless channel was considered with a maximum transmission capacity of 250 kbps [36] and a packet size of 80 bytes (value proposed by [10] as the value that best balances the latency and throughput). Then, according to (6), the maximum network load is reached with 390.62 packets per second (pps).
(6)$$\mathrm{Load}=250\mathrm{kbps}=\frac{31250\;\mathrm{bytes}}{\sec\mathrm{onds}}=390.62\;\mathrm{pps}$$

The test was designed to gradually increase the wireless channel load (from 0 to 400 pps) in order to measure the C, D_P_, and CAF values at each moment of node congestion. Therefore, each sensor node generates a constant bit rate (CBR) when it is active in the network. The test started with a single source node sending 50 pps over time. Subsequently, the number of active nodes was gradually increased until the maximum wireless channel limit was exceeded, with 20 nodes generating 400 pps or 256 kbps. This test was repeated 100 times for each parameter to increase accuracy and was integrated into a single dataset. These tests were developed with Network Simulator 2 (NS2) using the IEEE 802.15.4-2006 standard in non-beacon mode at 2.4 GHz, the UDP protocol, static routing, and an application agent CBR as a traffic generator.

The results of the simulation of these tests are shown in Figure 4, where the X1 axis (bottom) shows the evolution time and the duration of the simulation. The X2 axis (top) shows the evolution of the network load as the traffic flows are activated in each node, as time passes the network traffic increases. Where X1 and X2 are the independent variables in our experimentation. The Y1 axis (left) shows the percentage of packets lost in a node by CAF, and the Y2 axis (right) shows the degree of congestion C measured in the same node, which corresponds to the dependent variables. Another similar graph was obtained for the delay value D_P_. In this figure, it is necessary to establish two thresholds that divide the output data into three regions: the three congestion states. With this in mind, we propose that a low threshold (T_Low_) corresponds to a CAF value equal to or less than 1%, and a high threshold (T_High_) corresponds to a CAF value equal to or greater than 10%. This coincides with the moment when the output curves of the CAF and C exhibit abrupt changes in their results, which marks a trend in the node congestion degree. For the low threshold, when the CAF is equal to 1%, the value of the network load is 190 pps (equivalent to 47.5% of the total load), and the value of C is 1.012. For the high threshold, when the CAF was 10%, the network load was 285 pps (equal to 71.2% of the total load), and the value of C was 1.092. 

Table 1 specifies the thresholds’ values obtained in the tuning process for each congestion state.

Finally, we propose Algorithm 1 for node congestion degree detection. Our algorithm first checks the current congestion state of the node (the current state variable). Next, we review the current values of the congestion thresholds (C, CAF, and D_P_) to determine if it is necessary to move to a higher or lower congestion control state (variable congestion resolution). The tests carried out with this algorithm showed that it correctly selected the previously defined congestion states; therefore, the tuning process for the QCCP congestion-detection module was completed.
**Algorithm 1** Congestion detection functionData: C, D_P_, CAF, and Result: Congestion level: state1 **if** (current_state = = I) **then**2    **if** ((T_Low_ ≤D_P_ < T_High_) and (T_Low_ ≤ CAF < T_High_) AND (T_Low_ ≤ C < T_High_)) **then**3       current_state = II; congestion_resolution (state II);4    **else if** ((CAF ≥ T_High_) OR (C ≥ T_High_) OR (D_P_ ≥ T_High_)), **then**5       current_state = III; congestion_resolution (state III);6    **end**7 **else if** (current_state = = II) **then**8    **if** ((D_P_ ≥ T_High_) OR (CAF ≥ T_High_) OR (C ≥ T_High_)), **then**9         current_state = III; congestion_resolution (state III);10    **else if** ((D_P_ < T_Low_) AND (CAF < T_Low_) AND (C < T_Low_)) **then**11       current_state = I; congestion_resolution (state I);12    **end**13 **else if** (current_state = = III) then14    **if** ((T_L_ ≤ D_P_ < T_High_) AND (T_Low_ ≤ CAF < T_High_) AND (T_Low_ ≤ C < T_High_)) **then**15       current_state = II; congestion_resolution (state II);16    **else if** ((D_P_ < T_Low_) AND (CAF < T_Low_) AND (C < T_Low_)) **then**17       current_state = I; congestion_resolution (state I);18    **end**19 **end**

#### 4.2.3. Congestion Notification Mechanism

The congestion notification function has importance as a node needs the support of its neighborhood nodes to control congestion in the network. The literature mentions explicit and implicit notification as the ways in which a node can send control messages to its neighboring nodes. In the explicit form, the node must create a special packet to send the control information to other nodes. The advantages of the explicit form are that the node sends the packet only when needed and it has all the space in the packet to include the information. However, when the network is congested, sending this packet presents a problem, as it contributes to the congestion present in the network and the extra power consumption of the node. The implicit approach is where control information is piggybacked within the payload of the data packet itself. The implicit form does not have the problems of the explicit form, its only disadvantage is that it uses part of the payload of the data packet. Therefore, the least amount of information possible should be added to avoid creating a large overhead.

A design premise of QCCP is to control congestion regardless of the other nodes in the network. However, when the surrounding congestion causes a node to reach state III (high congestion), it sends an implicit notification to its neighboring nodes to work together to resolve the congestion. Specifically, QCCP takes one bit from the payload space of a data packet to indicate the presence (“1”) or absence (“0”) of high network congestion, as shown in Figure 5. This field is called high congestion notification (CN). The CN message is sent to its neighboring nodes taking advantage of the transmission nature of a common wireless channel [40] and the data reception function of the MAC sublayer of the node [36]. Thereby, first, the nodes take advantage of the shared network medium to receive packets from the neighborhood in their transmission range, regardless of the destination of each transmission, this is known as the overhearing effect [40]. Subsequently, when the node’s MAC sublayer receives a packet, it applies a series of filters (one of these is the verification of the destination MAC address) to decide whether to reject (dropped) or accept it (and pass it to higher layers) [36]. Before applying the filters, QCCP checks the value of the CN field, via the cross-layer connection, and submits the information to the congestion resolution mechanism to act accordingly. In conclusion, effective and fast notification is achieved without the need to send special packets to each node in the network and minimum overhead.

#### 4.2.4. Congestion Resolution Mechanism

Two strategies have been proposed to solve network congestion in a dynamic and decentralized manner at each network node: selective packet service and selective backoff. These strategies operate simultaneously and independently within the QCCP protocol according to its congestion detection mechanism and/or when a node receives a data packet with an extreme congestion notification (bit CN = “1”). The main objective of the congestion resolution mechanism is to resolve network congestion but guarantee that the packet processing delay (D_P_) and packet loss (CAF) are below a certain QoS threshold for the traffic’s highest priority (P1) during any congestion state. QCCP offers the best effort to the remaining traffic according to its priority through the selective configuration of the prioritization module. The two resolution strategies applied by the QCCP protocol are as follows.

##### Selective Packet Service

Selective packet service is a strategy based on node self-regulation to select the number (r_prog_) and types of packets that a node can transmit to a wireless channel. This process controls the CASPA algorithm (from the Prioritization module in Figure 3) to dynamically adjust the weights of each buffer (Q_1_, Q_2_, Q_3_, …, Q_n_) and determines which packets should be served and which should be queued on the transport layer. First, however, it must be ensured that the following rule is always fulfilled: Q_1_ > Q_2_ > Q_3_ …> Q_n_. This implies that more packets are always taken from the highest-priority traffic, as shown in Table 2, which presents the weighting proposal for the CASPA algorithm used in the tests. Each buffer obtains its weight according to value w (number of packets), which in turn depends on the storage capacity of the node.

As a special case, when a node reaches maximum congestion state III, we propose to stop transmitting packets of lower priority (Q_n_) to alleviate network congestion. It is preferable to drop low-priority packets by buffer overflow, rather than transmitting them to a channel with a high collision probability. This is because a packet lost by a collision consumes more energy from the node and unnecessarily increases surrounding congestion.

##### Selective BACKOFF

This strategy focuses on managing the node’s medium access algorithm to increase or decrease the data transmission capacity depending on the congestion state. In particular, we propose optimizing the CSMA-CA algorithm by setting the optimal value of the backoff exponent (BE) and the maximum number of backoffs (macMaxCSMABackoff) based on node congestion. This is necessary because various authors have shown that the default values for BE and macMaxCSMABackoff set by the IEEE 802.15.4 standard are not optimal for achieving the optimal operational performance of the algorithm [12,13,41,42]. It is necessary to understand the operation of the CSMA-CA algorithm. When the node MAC sublayer is ready to send a frame, CSMA-CA waits for a random period or backoff period (320 µs each) defined by (7), and then checks if the wireless channel is free to transmit.
(7)$$\mathrm{Backoff}\;\mathrm{period}=\left[0,\;2^{\mathrm{BE}}-1\right]$$

At the beginning of the process, the CSMA-CA algorithm assigns the value of the variable macMinBE to BE (defaults to 3) and increases its value by one (without exceeding the value of macMaxBE, defaults to 5) if it finds the channel busy. It then repeats the wait and check processes until the number of attempts to transmit the frame is reached (determined by the variable macMaxCSMABackoff by default four). Finally, if the frame cannot be transmitted, CSMA-CA declares a channel access failure (CAF) and drops it [36].

Based on a literature review, we propose a rule for managing the CSMA-CA algorithm by considering high values of BE and macMaxCSMABackoff to significantly reduce the packet loss probability if the packet transmission delay threshold is not exceeded. We replicate the tests in [39] with our network scenario to establish the optimal values of macMinBE, macMaxBE, and macMaxCSMABackoff (hereafter referred to as minBe, maxBE, and maxBackoff, respectively). The test results showed that the values proposed in [42], with some changes, are adequate for the effective operation of the CSMA-CA algorithm in different congested network states. The difference from our proposal lies in the way in which the three congestion states are defined, as the work [42] does not consider the effects of the transport layer (packet queuing effects) and uses other performance metrics to evaluate the algorithm efficiency (in our case, C, CAF, and D_P_ are used). The minBe, maxBE, and maxBackoff values for each congestion state are presented in Table 3.

Note that in States I and II, the values of minBE and maxBE are the same, which eliminates the possibility that the CSMA-CA algorithm modifies the BE value (7) every time the channel is busy. Thus, changes in the BE value that could affect the overall performance of the network are avoided. Furthermore, with these proposed values, we allow longer wait times (minBE and maxBE values) and a higher number of frame retransmission attempts (maxBackoff) to minimize packet loss probability while reaching an acceptable packet delay value. However, in the maximum-congestion state (State III), the packet delay values increased considerably. Therefore, we adjusted the values of minBE, maxBE, and maxBackoff to determine the best balance between packet loss and transmission delay. Coincidentally, the values found for State III are similar to those proposed by the IEEE 802.15.4 standard, except that we increase the maxBackoff value from 4 to 5, which reduces the packet loss percentage by the saturated channel.

Finally, the values of these two strategies, the weighting of the buffers Q_1_, Q_2_, and Q_3_ (Table 2), and the values of the CSMA-CA algorithm (Table 3) must be integrated into Algorithm 1 so that QCCP can control the congestion of the network and, at the same time, offers packet prioritization.

### 4.3. QCCP Protocol Parameters

Figure 6 shows the configuration values of the congestion detection and resolution mechanisms obtained in the tuning processes. We observed how the T_High_ and T_Low_ thresholds are used to pass from one congested state to another. For example, if a node is in medium congestion (State II), to move to State I, the values must be C < 1.01, CAF < 1%, and D_P_ < 11 ms. However, if the values are above C ≥ 1.1, CAF ≥ 10%, and D_P_ ≥ 22 ms, then the congested state transitions to State III. The CAF and D_P_ values are configured according to the QoS requirements of the ECG signals [37,43,44]. In the case of congestion resolution, on the one hand, the weights used by the CASPA algorithm of the prioritization module are shown (Q_1_, Q_2,_ and Q_3_). On the other hand, the minBE, maxBE, and maxBackoff values of the CSMA-CA algorithm are shown for each congestion state, which offered the best balance between network load, packet loss, and packet transmission delay. In addition, it is shown that in State III, the critical congestion notification bit (red box) is activated to inform the neighboring nodes, which can contribute to resolving congestion.

In conclusion, with this scheme executed by the QCCP protocol, we can achieve traffic prioritization and dynamic congestion control, which are aware of the degree of congestion and QoS needs of each packet. In addition, the QCCP protocol can be adapted to other network scenarios, because it allows the values of its parameters to be modified at any time from the transport layer.

## 5. Experimental Setup

This section describes the network configuration, traffic characteristics, and performance metrics used in the QCCP protocol evaluation.

### 5.1. Traffic Characteristics

The overall design considered a heterogeneous IoMT application scenario focused on monitoring remote patient vital signs. The network used in the tests consisted of 21 IoMT nodes distributed over an area of 35.0 × 35.0 m and an inter-node distance of 8.5 m, which ensured that all nodes were in the same wireless coverage (Figure 2). The IoMT has a single node that receives all network traffic, called a sink, and is located at the center of the coverage area.

Each node is equipped with sensors that can generate three types of physiological signals, an ECG signal (P1 traffic), a heart rate signal (P2 traffic), and a body temperature signal (P3 traffic), as explained in Section 3.1. The ECG signal is considered the traffic with the highest priority owing to its high communication requirements [37]. Each source node uses a CBR-type application layer traffic generator, where each packet has a size of 80 bytes, according to [10] (Section 3.2). For our experiments, each source node was configured to generate a data rate according to the QoS requirements of each physiological signal, in tune with [37]. We modified the CBR application to add a priority label to each packet, according to the previous classification.

An experiment was designed such that the traffic load on the network had a bell-shaped curve to evaluate the performance of QCCP in each state of congestion (low, medium, and high). The top of the curve corresponds to the maximum saturation of the IEEE 802.15.4 wireless channel (250 kbps) [36]. Network traffic begins with a single node generating 50–150 pps for a particular time (State I). Subsequently, the number of nodes gradually increased until reaching the saturation point, State III, generating a network load of 400 pps (or 256 kbps), which exceeded the wireless channel capacity. Finally, an inverse operation is applied to return to the curve base (State I). This design allowed us to measure the QCCP performance at different loads and the response speed when the network load was decreased.

Finally, the experiments considered the constraints of devices based on the IEEE 802.15.4 standard, characterized as the most restrictive in terms of computing resources, such as memory, power supply, and bandwidth (see Table 4). The rest of the configurations used in the QCCP protocol tests are shown in Table 4.

### 5.2. Performance Metrics

We measured the effects of network load on QoS parameters to obtain the performance of the QCCP protocol. Therefore, we propose three performance metrics to observe the results offered by the protocol against network congestion [32,34,45].

#### 5.2.1. Packet Transmission Latency (L)

*L* measures the time taken to travel a packet from a source node to the sink and is measured in the transport layer. This metric involves queue, processing, and medium access delays to achieve an end-to-end transmission.

#### 5.2.2. Packet Loss Percentage (P_L_)

P_L_ measures the percentage of packets lost by network nodes during a time interval. The losses due to buffer overflow, channel access failure, and packet collisions are included in the P_L_ value. The P_L_ value was obtained by dividing the total number of packets lost in the test by the total number of packets generated by the nodes in the application layer.

#### 5.2.3. Throughput (T_H_)

T_H_ measures the total percentage of packets successfully received by the sink over time.

L, P_L_, and T_H_ metrics were measured for each node and type of traffic (P1, P2, and P3) to measure the network’s individual performance. Each simulation consisted of 800 s of testing. All the results are shown after averaging the metrics over 100 different test runs. The NS2 simulator was configured with a randomness factor that caused the nondeterministic operation of the network in each simulation. Emphasis is placed on the operation of the CSMA-CA algorithm (to select the backoff periods) and the behavior of the wireless channel (datasets are available upon request).

## 6. Experimental Study

Experiments were conducted using two case studies. The first case evaluates the performance of QCCP for each type of traffic within the test scenario. The second case compares the performance of QCCP with other well-known transport protocols (TCP): Tahoe (the base version of TCP) [46], Newreno [47], and Vegas [48], with the primary objective of evidencing the operation of QCCP rather than competing with them. The TCP protocol was used in the tests because it is currently the most common way to complete an internet connectivity stack. Moreover, TCP is used as the basis of the message queuing telemetry transport (MQTT) protocol, which is one of the main proposals for IoT; since MQTT needs a protocol that provides orderly and lossless bidirectional connections such as TCP [49].

### 6.1. QCCP Performance Evaluation for Each Type of Traffic (First Case)

The results of the first case study showed that QCCP guaranteed the best performances in the three metrics (L, P_L_, and T_H_) according to the priority of each traffic against the network load (Figure 7, Figure 8 and Figure 9). For example, Figure 7 shows the average packet transmission latency (L) values, from node 1 to the sink node, for each type of traffic. The “urgent (P1) traffic” obtains the best latency values (less than 20 ms) throughout the entire simulation, the same as “Important (P2) traffic”; while it offers the best option for P3 traffic in medium loads (States I and II). This is because, in the presence of critical congestion, QCCP keeps less important packets in the queue for a longer period, which increases the packet transmission latency. 

As a result of this queuing, as the buffer reaches its storage limit, the packets are dropped to control network congestion. In this regard, Figure 8 shows the number of packets lost (both due to buffer overflow and channel saturation) in the network for each type of traffic. For example, in the same figure, when the network is at maximum congestion (400 pps—axis X2), the network loses 81.1% for P3 traffic, 33.5% for P2 traffic, and 27.6% for P1 traffic. This can be drastic for P2 and P3 traffic; however, it is necessary to eliminate network congestion to support the network QoS requirements. Furthermore, the largest number of packets are dropped internally in the nodes, so it does not contribute to the congestion of the wireless channel. Finally, we highlight that in low and medium congestion, QCCP reaches values below 1% and 10% for P1 traffic (at loads of 150 pps and 220 pps on the X2 axis), respectively, fulfilling the expected performance in the established T_High_ and T_Low_ thresholds.

Figure 9 shows how the QCCP manages data transport in the network (T_H_) according to the QoS requirements and network congestion state. For example, when the network congestion is low (State I), the throughput value is balanced for all types of traffic. However, as the network congestion increases, QCCP offers performance based on the priority of each traffic, as shown in the middle of Figure 8 (325 pps of the X2 axis), where the maximum network congestion starts (State III). As expected, the highest channel throughput was obtained for essential traffic (P1), followed by non-real-time traffic (P2). Leaving the lowest-priority traffic (P3: best effort) with the lowest channel throughput. In State III at 400 pps, on axis X2, the value of T_H_ reached by traffic P1 is 24.55% and 86.50% higher than traffic P2 and P3, respectively.

These results demonstrate that the QCCP protocol can prioritize all types of traffic while resolving network congestion.

### 6.2. QCCP Performance Comparison with Other Transport Protocols (Second Case)

The second set of simulations was conducted to compare the performance of QCCP with that of some major TCPs protocols, and the results are shown in Figure 10, Figure 11 and Figure 12. 

In general, QCCP outperformed the TCP protocols for all evaluation metrics. The results are as follows.

Figure 10 shows the average transmission latency for P1 traffic in the network, where the worst results were obtained by Tahoe and Newreno, exceeding 50 ms. Meanwhile, Vegas performs well, with a latency value of around 28 ms. However, the best results were obtained with QCCP, below 22 ms. For example, with a load of 400 pps (axis X2) latency values of 28.5 ms and 14.9 ms were achieved by Vegas and QCCP, respectively. This result represents a 47.87% improvement in QCCP.

Figure 11 shows the average network packet loss (P_L_) for P1 traffic. The results show that QCCP outperforms TCP protocols, except in the maximum congestion range (between 320 and 400 pps on axis X2), where Tahoe and Newreno obtain fewer lost packets than QCCP. However, this improvement is relative because the QCCP transmits more packets per second to the sink node than under the same conditions (Figure 11). Furthermore, one of the most important results observed in Figure 10 is that QCCP reacts appropriately to network congestion conditions by adequately adjusting the congestion window, whereas the TCP protocol does not. For example, even though the network load begins to decrease considerably (by 185 pps, X2 axis), the TCP protocols still lose a large packet percentage (above 25%).

Figure 12 shows the average throughput on the network for P1 traffic, and it is observed that QCCP outperformed the TCP protocols in almost the entire test period. For example, in State III (400 pps of the X2 axis), the QCCP protocol reaches an average T_H_ value of 87.38 pps, Vegas 31.18 pps, Newreno 29.10 pps, and Tahoe 28.66 pps. Improvements of 64.31% and 66.69% were observed for Vegas and Newreno, respectively. These results are twofold. First, QCCP can better identify network congestion, and secondly, apply a smooth adjustment of the number and type of packets that every node can send to the wireless channel, unlike TCP protocols. Confirming that the QCCP protocol achieves excellent QoS levels in highly congested networks.

### 6.3. Summary of the Experimental Study

QCCP performance was analyzed based on two major objectives: controlling congestion based on the current network load and packet prioritization, as seen in Figure 7, Figure 8 and Figure 9, as was established in the first case (Section 6.1).

Table 5 summarizes the results from the experiments for the second case, which shows how QCCP outperforms TCP protocols in almost all scenarios. We compared the results with other well-known transport protocols such as Tahoe, Newreno, and Vegas, as shown in Figure 10, Figure 11 and Figure 12. The results showed that QCCP offers better QoS than the TCP protocols. Unlike TCP protocols with flat and aggressive control, QCCP offers a smooth adjustment of the amount and type of traffic.

## 7. Statistical Analysis

A statistical analysis of the evaluated protocols was presented to demonstrate the validity of the obtained results. To define which statistical test should be used, it is necessary to verify whether the datasets follow a normal distribution. For this purpose, three datasets corresponding to each protocol’s metrics T_H_, L, and P_L_ were used. Each dataset comprised 100 samples obtained from a hundred network simulations. These samples of each dataset correspond to the exact moment where the IoMT network presents the highest traffic congestion (400 pps or 256 kbps), seeking to validate the operation of the protocols in the “worst situation”. The dataset files are available to the community upon request.

### 7.1. Data Normality Test

The Kolmogorov–Smirnov test with Lilliefors correction was used, which is a well-accepted tool to prove data normality. In this test, there is a null hypothesis (H_0_) that states that the data follow a normal distribution if the significance value (*p*-value) is greater than 0.05 (which is equivalent to 95% confidence). Otherwise, they are rejected. The results obtained from the normality tests for the three datasets are presented below. The QCCP protocol dataset showed that the data followed a normal distribution because *p*-values of 0.1571, 0.1126, and 0.06869 were obtained for the L, P_L_, and T_H_ metrics, respectively. However, the Tahoe, Newreno, and Vegas datasets did not exhibit a normal distribution in all cases. For example, all TCP protocols satisfy the normality assumption for the P_L_ metric dataset. However, in the L metric, no dataset complied with data normality (Newreno with *p*-value = 8.729 × 10^−7^, Tahoe with *p*-value = 2.488 × 10^−9^, and Vegas with *p*-value = 0.02409).

### 7.2. Non-Parametric Tests

From the previous section, we know that not all datasets follow a normal distribution, so to prove that QCCP is statistically different from the TCP protocols, we must perform non-parametric tests. Therefore, we decided to use Wilcoxon’s statistical test for paired samples because the four transport protocols are executed in the same network scenario, which means that there are no changes except for the protocol. Therefore, this raises the question: Are there any significant changes in network performance values with the execution of any particular protocol?

In the Wilcoxon test, the null hypothesis (H_0_) specifies that there is no difference between medians (Md); therefore, the protocols analyzed are statistically equal (Md_QCCP_ = Md_TCPs_), meaning that the network’s overall performance will be almost the same regardless of which protocol is executed. Where Md_QCCP_ denotes the median value of the QCCP protocol. Md_TCPs_ denotes the median value for the TCPs = Vegas, Newreno, and Tahoe protocols. Therefore, we are looking for a *p*-value ≤ 0.05 (α), which rejects H_0_, meaning that the protocols analyzed are statistically different (H_1_; alternative hypothesis), opening a window to the possibility that QCCP outperforms the other protocols. For consistency, we used the Wilcoxon test for latency, packet loss, and throughput metrics to examine whether QCCP was statistically different from the other TCP protocols. For the statistical tests, the R program and Wilcoxon signed-rank test function were used. The latter is used for sample sizes greater than 25, which also returns the exact *p*-value value instead of an approximation. 

Table 6 presents the *p*-values obtained for each protocol and a performance metric comparison. The *p*-value of every one of each test is less than the significance level alpha = 0.05. Therefore, we can reject the null hypothesis and confirm that there is a statistical difference in the results obtained by each protocol (QCCP, Newreno, Tahoe, and Vegas protocols). In other words, these tests allow us to conclude that the performance value of each metric significantly depends on the transport protocol running on the network.

### 7.3. Statistically Performance Validation of QCCP

In this section, we describe an inferential statistical analysis of the results obtained in QCCP to test the hypothesis that the average of its performance metrics (L, P_L_, and T_H_) is in the range of transmitting an ECG signal when the network is in critical congestion, that is, when the network load has traffic of 400 pps. With this goal in mind, the average values required to transmit ECG signals over the network are described below.

According to [37], the average T_H_ of QCCP must be greater than 15,000 bps or 23.44 pps (80 bytes packet size) to transmit a 9-lead ECG signal. Furthermore, in [50], it was established that an ECG signal must be sent with a latency of less than 500 ms. However, because our dataset records the latency value per packet, this value changes to 21.33 ms (500 ms divided by 23.44 packets), which is the limit for delivering a packet to the sink node. In the case of the packet loss metric (P_L_), some studies have shown that a continuous transmission of ECG data for 30 s, with up to 8% packet loss, can be functional for medical interpretation [51]. Therefore, in our experiment, where the source nodes generate 400 pps at maximum congestion in 30 s of transmission, there would be a total of 12,000 packets, of which 960 packets (8%) are allowed to be lost in 30 s or 32 packets in one second. In conclusion, the hypotheses of the statistical test are for T_H_, µ_TH_ = 23.44 pps (H_0_) and µ_TH_ > 23.44 pps (H_1_); for L, µ_L_ = 21.33 ms (H_0_) and µ_L_ < 21.33 ms (H_1_); and for P_L_, µ_PL_ = 32 packets (H_0_) and µ_PL_ < 32 packets (H_1_). µ_r_ denotes the mean value for r = T_H_, L, and P_L_.

Regarding the three metrics mentioned above, their datasets follow a normal distribution (Section 7.1); therefore, we used the Student’s *t*-test to carry out the inferential test, using an R program with α = 0.05, and consequently, a confidence level of 95%. Table 7 presents the results of the test.

As shown in Table 7, the *p*-values were less than 0.05. Therefore, the null hypothesis is rejected for each metric. For example, for the case of the L metric, Table 7 shows that the QCCP protocol achieves a lower average latency of 0.01483 s than the target average value (0.0213 s), statistically demonstrating that QCCP can support the indicated QoS-latency value. In Table 7, the limit value corresponds to the upper level of the confidence interval for the T_H_ metric, whereas the other two metrics (L and P_L_) correspond to the lower level of the confidence interval.

Therefore, with a 95% confidence level, the QCCP protocol is suitable for transmitting ECG signals in an IoMT with traffic congestion problems.

## 8. Conclusions

In this article, a congestion control protocol was presented with the purpose of offering traffic prioritization in congested IoMT networks, called QCCP. QCCP is made up of packet prioritization and congestion control modules, working together to prioritize traffic, and detect, notify, and resolve network congestion.

After carrying out the protocol simulations and verifying them with statistical analysis, it is concluded that QCCP works better than the compared protocols. In addition, the statistical analysis allowed us to verify the hypothesis that QCCP is suitable for the transport of ECG signals (type 1 traffic) in a congested sensor network.

Thus, the proposed solution can offer traffic priority while achieving adequate throughput, low latency, and low packet loss in highly congested networks. Consequently, it can significantly benefit IoMT technology by saving time, cost, and more importantly, human lives. This drives our future work to implement the QCCP in physical devices to demonstrate its excellent benefits.

## Figures and Tables

**Figure 1 sensors-23-00923-f001:**
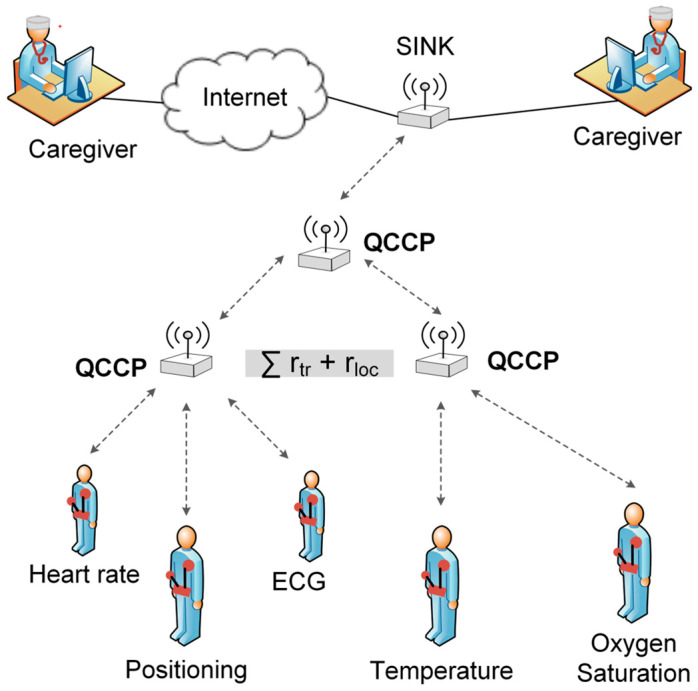
IoMT healthcare application scenario.

**Figure 2 sensors-23-00923-f002:**
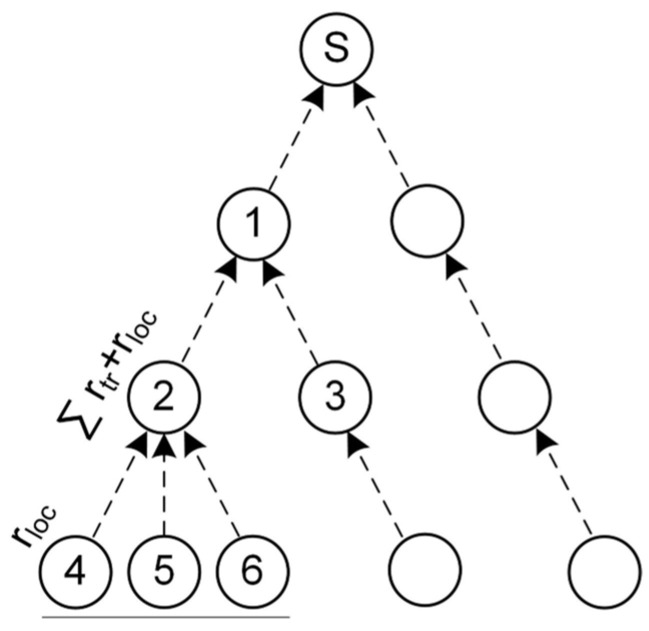
Network model.

**Figure 3 sensors-23-00923-f003:**
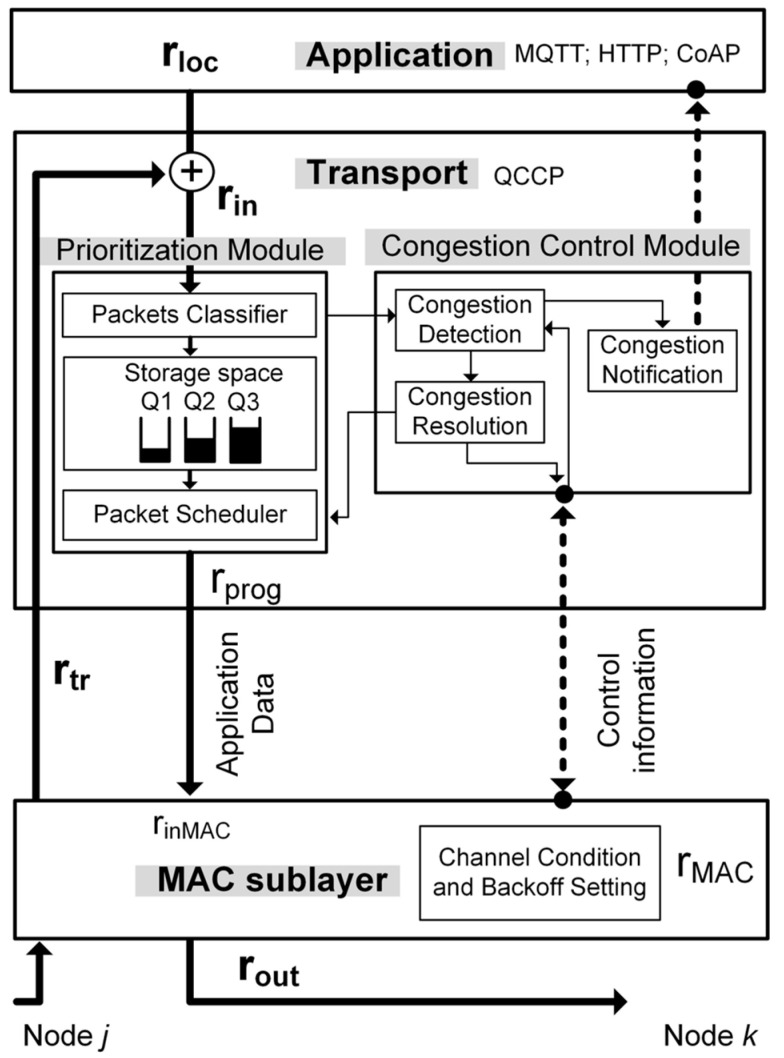
Node model and cross-layer interaction.

**Figure 4 sensors-23-00923-f004:**
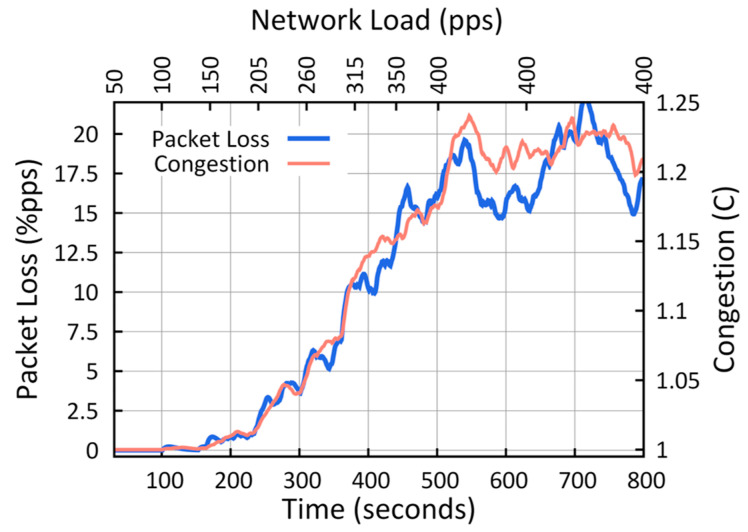
Values of CAF and C in the node versus network load.

**Figure 5 sensors-23-00923-f005:**
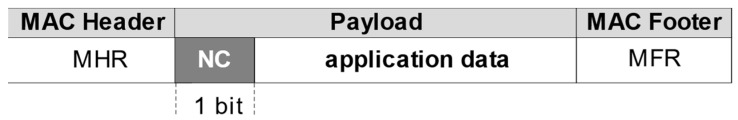
Proposed modification of the data frame for congestion notification.

**Figure 6 sensors-23-00923-f006:**
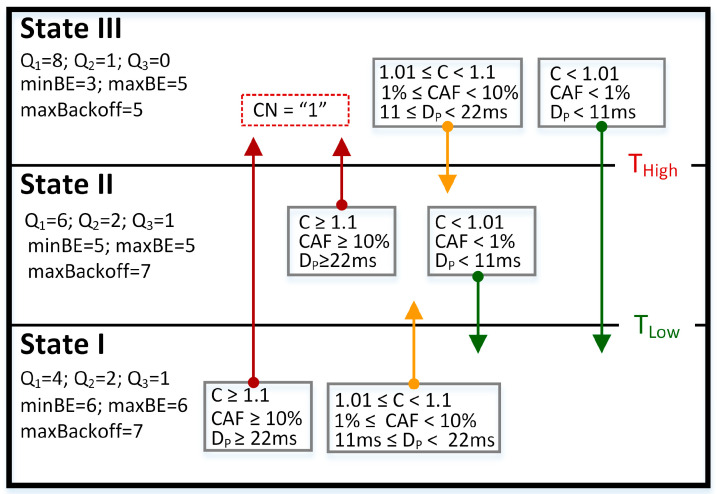
Key values for the operation of QCCP.

**Figure 7 sensors-23-00923-f007:**
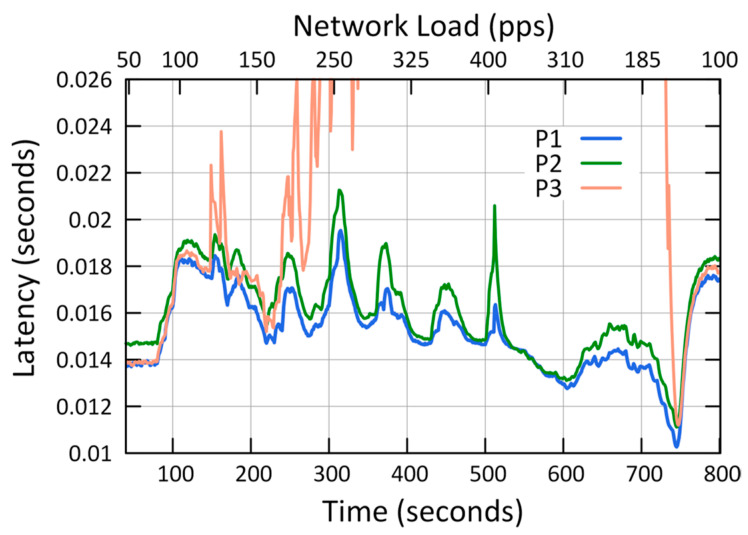
Transmission latency (L) of node 1.

**Figure 8 sensors-23-00923-f008:**
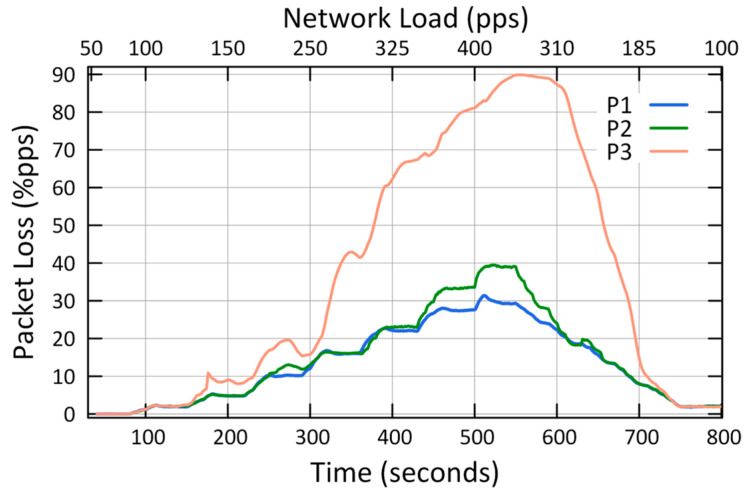
Packet loss (P_L_) in the network.

**Figure 9 sensors-23-00923-f009:**
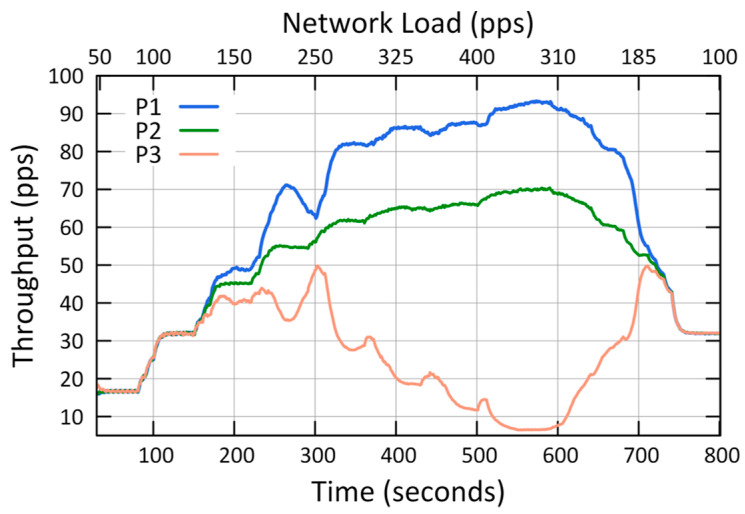
Throughput (T_H_) in the Network.

**Figure 10 sensors-23-00923-f010:**
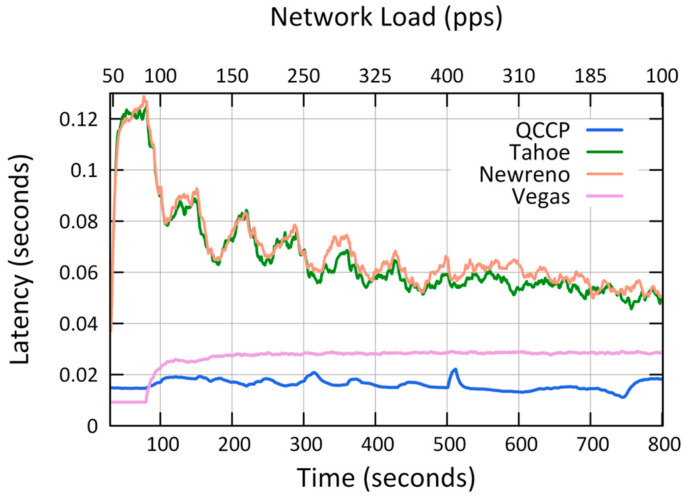
Comparison of average packet transmission latency (L).

**Figure 11 sensors-23-00923-f011:**
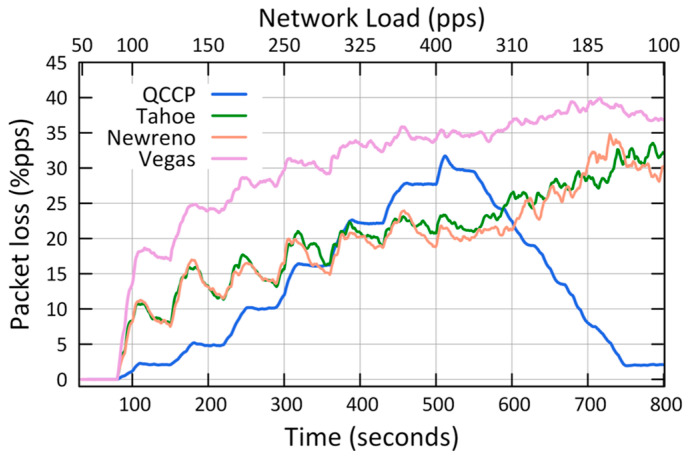
Comparison of average packet loss (P_L_).

**Figure 12 sensors-23-00923-f012:**
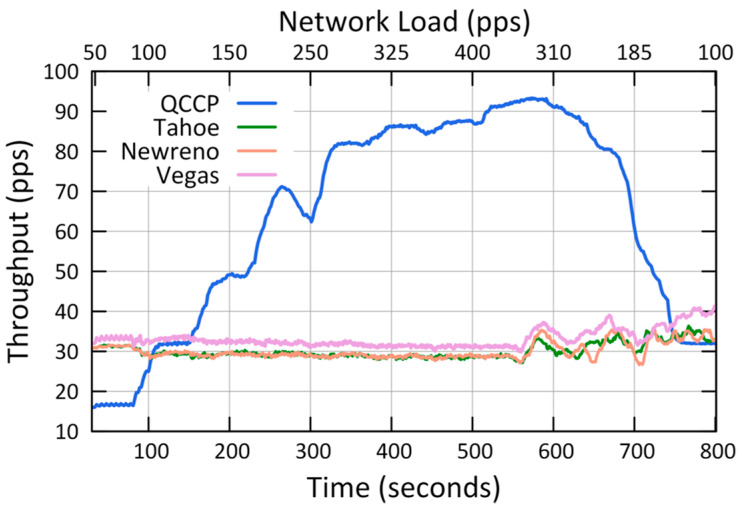
Comparison of average throughput in the network (T_H_).

**Table 1 sensors-23-00923-t001:** Congestion state and thresholds.

State I	State II	State III
D_P_ < 11 ms	11 ms ≤ D_P_ < 22 ms	D_P_ ≥ 22 ms
CAF < 1%	1% ≤ CAF < 10%	CAF ≥ 10%
C < 1.01	1.01 ≤ C < 1.1	C ≥ 1.1

**Table 2 sensors-23-00923-t002:** CASPA algorithm weighting proposal.

Congestion	Q_1_ Buffer	Q_2_ Buffer	Q_3_ Buffer
State I	Weight = w × 1.0	Weight = w × 0.5	Weight = w × 0.25
State II	Weight = w × 1.5	Weight = w × 0.5	Weight = w × 0.25
State III	Weight = w × 2.0	Weight = w × 0.25	Weight = w × 0

**Table 3 sensors-23-00923-t003:** Proposed values for the CSMA-CA algorithm.

Congestion	minBE	maxBE	maxBackoff
State I	6	6	7
State II	5	5	7
State III	3	5	5

**Table 4 sensors-23-00923-t004:** Setting parameters for network and node.

Parameter	Values
Application agent	Constant Bit Rate (CBR)
Routing agent	Static routing
Transport protocols	QCCP, TCP (Tahoe, NewReno, Vegas)
MAC-PHY protocol	IEEE 802.15.4 (Non-Beacon mode)
Node’s size buffer	seven packets for each queue
Packet size	80 Bytes [10]
Wireless channel rate	250 kbps (one channel at 2.4 GHz)

**Table 5 sensors-23-00923-t005:** Experimental result summary of second case.

Performance Metric	State I 190 pps	State II 285 pps	State III 400 pps
L (seconds)			
Tahoe	0.0736	0.0686	0.0588
Newreno	0.0771	0.0745	0.0655
Vegas	0.0275	0.0282	0.0286
QCCP	0.0162	0.0161	0.0149
P_L_(% pps)			
Tahoe	13.7433	16.4479	20.7432
Newreno	13.1389	14.8692	18.8666
Vegas	25.3355	29.2678	34.0296
QCCP	6.3170	16.2162	27.6790
T_H_ (pps)			
Tahoe	29.5580	29.2245	28.6615
Newreno	29.3666	29.3547	29.1040
Vegas	32.8587	31.5950	31.1818
QCCP	52.1215	81.9238	87.3884

**Table 6 sensors-23-00923-t006:** Wilcoxon test results.

Metric	Compared Protocols		Statistic W	*p*-Value
	QCCP (Md)	Newreno (Md)		
T_H_	86	28.50	8.68	<0.001
L	0.0147	0.0535	−8.68	<0.001
P_L_	27.70	18.58	7.13	<0.001
	QCCP (Md)	Tahoe (Md)		
T_H_	86	30	8.68	<0.001
L	0.0147	0.0430	−8.68	<0.001
P_L_	27.70	19.70	6.41	<0.001
	QCCP (Md)	Vegas (Md)		
T_H_	86	31.50	8.68	<0.001
L	0.0147	0.0274	−8.68	<0.001
P_L_	27.70	33.80	−7.60	<0.001

**Table 7 sensors-23-00923-t007:** Statistical analysis of QCCP performance, with a simulation window of 800 s.

Metric	Mean Value	Limit Value	T-Value (GL = 99)	*p*-Value (α = 0.05)	Target Value
T_H_	86.47	85.460	103.62	<0.001	23.44
P_L_	27.62	27.85	−30.55	<0.001	32
L	0.01483	0.0150	−59.4	<0.001	0.0213

## Data Availability

The data presented in this study are available on request from the corresponding author.

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
