# Peer review of "Prioritization-Driven Congestion Control in Networks for the Internet of Medical Things: A Cross-Layer Proposal"

_sensors, 2023, doi:10.3390/s23020923_

Round 1

Reviewer 1 Report

The paper studies the prioritization and congestion issues in IoT networks, which lead to increasing packet loss likelihood, latency and high-power consumption in healthcare systems. This are important issues in the real-life implementation of IoMT. Motivated by the limitations of the existing protocols, the study proposes a novel prioritization-based cross-layer congestion control protocol called QCCP. QCCP estimated the congestion degree of the wireless channel from the internal node congestion state, without the need to exchange information with other nodes in the network. QCCP processes all tasks using four significant functions: packet prioritization, congestion detection, congestion notification and congestion resolution.

However, there are several weaknesses in this paper. Improvement is expected if the authors can take the following points in to account.

1)     The authors proposed a network model in section 3.1 to analyze the IoMT nodes. However, the concepts such as sink node, offspring node, and child node are not well defined mathematically. In addition, the process of transferring data among the nodes need more explanation to make it clearer for readers.

2)     The authors may want to give more details about the congestion notification mechanism. Use equations or figures to present how this mechanism works.

3)     The experiment analysis in the Conclusion should be moved the experiment result discussion part. The authors may want to summary the whole paper in Conclusion section.

4)The authors failed to cite several past literatures highly related to this work (e.g., [1-3]) and clearly discuss the differences between them and this paper.

[1] PCC Vivace: Online-Learning Congestion Control. NSDI 2018

[2] MIRAS: Model-based Reinforcement Learning for Microservice Resource Allocation over Scientific Workflows, ICDCS 2019. 

[3]  Multi-objective congestion control. EuroSys 2022:

5)     Some figures are not clearer and need modification, including fig 5, fig 6. In addition, the axis meaning of fig 4 and fig 6 – 10 should be explained to make them easier to understand.

6)     In fig 6, the authors may alter the “p1, p2, p3” with more concrete description to show the different settings among the three curves.

7)     The section 4.2.2 parameter tuning may need to be moved to the experiment part.

8)     The pseudo code of algorithm 1 need to be brief. The authors may want to explain why it is called a function, since it is written in form of algorithm.

9)     In section 4.1, the authors may want to give a pseudo code of the proposed CASPA algorithm.

Overall, this is a solid piece of work, but major modification is needed.

Reviewer 2 Report

The goal of this paper, as exposed by the authors, is to present QCCP protocol, which is managed by communication devices' transport and MAC layers.

Please Quantify the main results in the introduction and conclusion sections, compared to other authors' articles.

Related work is too short and concise. In order for the article to be accepted for publication, this chapter must be significantly improved both in terms of content size and quality. Figure 1 needs to be improved by adding additional information (OSI layers), and figure 2 can be explained in more detail. On which OSI layer is QCCP implemented (the content of Figure 3 is not very visible and this information is specified too late (in Figure 3))? What functions are implemented compared to other congestion control protocols (SACC, HOCA, ..)?

QCCP is implemented at the transport level, being represented only in the end systems. Routers and switches implement OSI layers 2 and 3. Do these systems have access to layer 4 (transport)? Are they in accordance with the proposed protocol? Figures 3 and 5 are not in good resolution, so they must be redone. The authors must consider additional experimental results situations based on traffic congestion, hardware&software constraints or different QoS scenarios.

The comparison regarding the results obtained for QCCP, Tahoe, Vegas and Newreno are useful and well presented. The authors should clearly the personal contributions for THIS paper compared to previous research papers.

The reference section is good, citing new and relevant articles in the research area.

Reviewer 3 Report

In this revised manuscript draft, the authors has proposed a priority-based cross-layer congestion control protocol called QCCP. The experiments show that QCCP outperforms other TCP protocols, with 64.31% higher throughput, 18.66% less packet loss, and 47.87% less latency.

However, this paper is not flawless. My detailed comments are as follows:

1. There are a lot of abbreviations (such as SL, SCN) in the article, please summarize them on a table.

2. The pictures in Fig. 5 and Fig. 6 are not clear, please replace them wtih eps format.

3. Enrich the content of related works, for example, introduce the comparison algorithms.

4. The comparison algorithms used in the experiment are not the latest algorithm, and most of them were published before 2004. Newest researches can be added to show the novelty of the experiments.

Round 2

Reviewer 1 Report

comments addressed. recommend acceptance

Reviewer 2 Report

The paper was improved by the revision process.